# Factors associated with viral load suppression among orphans and vulnerable children and adolescents living with HIV in Kenya

**Brian Onyango**[1]*, **Rose Mokaya**[1], **Jeniffer Wasianga**[1], **Hesborn Wao**[2],
**Dunstan Achwoka**[1], **Nelson Onyango**[3], **Damazo T. Kadengye**[2]

**1** United States Agency for International Development (USAID), Nairobi, Kenya, **2** African Population and Health Research Center (APHRC), Nairobi, Kenya, **3** School of Mathematics, University of Nairobi, Nairobi, Kenya

* BOnyango@usaid.gov

**Data Availability Statement:** All relevant data are within the paper.

## Abstract

While tremendous progress has been made on attaining HIV treatment goals (95-95-95), children's viral load suppression remains a challenge particularly among the orphans and vulnerable groups. In Sub Saharan Africa, there is limited evidence of specific interventions in orphans and vulnerable children (OVC) programs to support children and adolescents living with HIV (CALHIV) to attain durable viral load suppression. Through a large OVC cohort, the study sought to identify correlates of optimal viral load suppression among CALHIV in the Kenya OVC program. This cross-sectional study utilized data on CALHIV below the age of 18 years who were enrolled in the OVC program and actively receiving HIV care and treatment services from ART clinics across Kenya and with documented VL results between October 2019 and September 2020. To obtain a nationally representative sample, data was retrieved from USAID implementing partners' databases across the country. Association between selected variables and VL suppression (outcome of interest) were assessed using a multivariate mixed effect logistic regression model, using *glmer* function in the LME4 package in R. Factors associated with VL suppression included child's education status (aOR = 1.33; 95% CI: 1.07, 1.65), membership of a psychosocial support group (aOR = 1.258; 95% CI: 1.15, 1.38), and membership of a voluntary savings and lending association (VSLA) (aOR = 1.226; 95% CI: 1.129, 1.33). In addition, child's sex (aOR = 0.88; 95% CI: 0.83, 0.94), caregiver sex (aOR = 0.909; 95% CI: 0.839, 0.997) and "high" status for caregiver household vulnerability (aOR = 0.81; 95% CI: 0.71, 0.924), had an inverse relationship with VL suppression. CALHIV characteristics including child's sex, child's education status (whether currently active in school or inactive) and child's membership in a psychosocial support group were key determinants of VL suppression. Similarly, caregiver sex and membership in a voluntary savings and lending association also influence VL suppression.

**Funding:** The authors received no specific funding for this work.

**Competing interests:** The authors have declared that no competing interests exist.

## Introduction

HIV remains one of the leading causes of mortality in children and adolescents in Africa [1]. In the 2014 report titled *Fast Track*: *Ending the AIDS epidemic by 2030*, the Joint United Nations Programme on HIV/AIDS (UNAIDS) the 95–95–95 treatment targets were set with the aim of transforming the vision of zero new HIV infections, zero discrimination and zero AIDS-related deaths into concrete milestones and endpoints. Despite tremendous progress, children living with HIV are disproportionately affected and therefore lagging in the achievement of these targets [2, 3].

Of approximately 38 million people living with HIV worldwide in 2020, 2.78 million were children and adolescents aged 0–19 years. About 850 children contracted HIV, and about 330 children died from AIDS-related causes in 2020. Additionally, of the estimated 680,000 people who died from AIDS-related diseases in 2020, approximately 18 percent were children under the age of 20. This can be mainly attributed to insufficient access to HIV prevention, care, and treatment services [4].

By 2020, close to 15.4 million children under the age of 18 had lost one or both parents to AIDS-related causes worldwide [4]. Kenya alone has an estimated 2.6 million orphans and vulnerable children (OVC) of whom 650,000 are orphaned due to AIDS [5]. Millions of people have been affected by the epidemic due to an increased risk of poverty, homelessness, school drop-out, discrimination and missed opportunities, as well as COVID19. These adversities are protracted illness and death. While deliberate steps have been made to support OVC, specific interventions for children living with HIV are nascent. Structural interventions that include social safety net initiatives and schemes are generic. Development partners have introduced elements of children and adolescents living with HIV (CALHIV) specific support albeit in a limited way.

Data on viral suppression (VS) among HIV positive children and adolescents receiving antiretroviral therapy (ART) as part of routine care in resource-constrained settings is not readily available [6]. However, available studies point to a suppression rate below 75%, way below the set 90% target [1, 7, 8].

Adherence to ART is a complex process influenced by a variety of factors. Several reasons cited for poor adherence to ART include financial instability, poverty and food insecurity, limited access to medical resources, and costs of transportation. Good adherence and retention are noted to be necessary requirements for achieving viral suppression. Adolescents are particularly vulnerable to household poverty, yet adherence to treatment regimens requires a level of economic stability that many poor youths in Africa do not have or experience [1, 9]. Several publications serve to underscore our inferences. These include a randomised control study in Uganda by Ssewamala et al who measured the effect of a five-year family economic empowerment intervention among Ugandan OVC families. Viral Load suppression rates increased and transmission rates among young people declined [1]. A South African study documented how lack of financial support led to a negative outcome for ART adherence. Additionally, low-income security within low-income households had a negative effect on adherence [10]. Finally, Nasuuna and colleagues highlighted the need for support with disclosure of HIV status to children and families and linkage to support groups, parenting classes and economic opportunities for caregivers to support adherence especially for the unsuppressed children [11].

Van Wyk and colleagues describe some of the structural barriers to care that ultimately affect adherence. For instance, poor service delivery, missing or misplaced files and long waiting times. Additionally, in many low resourced settings, facilities are far apart, and transport costs are expensive with few transport options. Long wait times are a turn off to clients who may be required to walk long distances or must contend with restricted options for travel.

Long wait times are correlated with missed appointments and loss to follow up. Caregiver training is essential for improved adherence to timing and correct dosing of medication. It further facilitates assisted disclosure thus assuring improved outcomes among OVC [10].

By identifying correlates of optimal viral load suppression amongst CALHIV in a large OVC program, we seek to add to the growing literature on social determinants of OVC health and address policy gaps to accelerate viral load suppression in low resource settings.

The conceptual framework underpinning this study was adapted from the socio-ecological framework that depicts layers of individual, relationship, community, and healthcare policy level factors that influence treatment adherence and retention in care [12]. This study will examine individual-level factors such as age, sex, household income and education as well as interpersonal factors such as family members or caregivers and community factors such as distance to health facility and support services and how they correlate to optimal viral load suppression.

## Methods

### Ethics statement

The study utilised anonymized secondary programmatic monitoring data extracted from USAID local implementing partner projects of a large PEPFAR funded program in Kenya. No ethical approval was sought. However, administrative written permissions and concurrence to extract data from the partners Child Protection Management Information System databases were obtained from Government of Kenya (GOK) through the Department of Children Services (DCS) and USAID country office through the Health, Population and Nutrition Office and the Agreement Officer Representatives (AORs).

### Study setting

It is estimated that approximately 1.5 million people are living with HIV in Kenya of which 7% (105,200) are children under the age of 15 years and 184,700 (12%) are the youth aged 15–24 years [12]. CALHIV are spread across rural and urban counties in Kenya. This study focussed on the CALHIV population in 34 counties– 2 urban and 32 rural–as defined in the 2014 Kenya Demographic Health Survey [13].

This study utilised cross-sectional data of CALHIV aged 18 years and below, who were actively receiving HIV care and treatment services from ART clinics across Kenya and with documented viral load results between October 2019 and September 2020. To obtain a nationally representative sample, data of CALHIV supported by USAID implementing partners namely MWENDO, CASE OVC, AMPATHplus and COGRI Orphans and Vulnerable Children Projects from across the country was retrieved from partner databases (Child Protection Information Management System) for inclusion in the study. Data of CALHIV who were actively receiving services in the OVC program between October 2019 and September 2020, and who had documented viral load results were included in this study. In total, data for about 65,816 CALHIV was extracted of which 7,918 are in urban counties and 57,898 in rural counties.

### Data extraction and variables

In reference to PEPFAR OVC programmatic indicators, a structured data extraction tool was developed in Microsoft Excel version 2016. Secondary data was extracted from the OVC service delivery partners' databases (Child Protection Management Information System) through the support of the project-specific service delivery Monitoring and Evaluation (M&E) teams. The template captured socio-demographic characteristics such as age, gender, schooling status

information for both the OVC and caregiver information. Other relevant data that were extracted included: Socio-economic characteristics such as membership in Voluntary Savings and Lending Associations (VSLAs), kind of Income Generating Activity (IGA), and the estimated average monthly income were provided at household level; Case Plan Achievement Readiness Assessment (CPARA) to inform service provision and tracking graduation readiness from OVC program support. This includes benchmarks organised under the Healthy, Safe, Schooled, and Stable domains at household level; and lastly, health related information such as the latest documented viral load information, distance to health facility where the CALHIV is receiving treatment, registration into psychosocial support (PSS) group for both the OVC and caregiver and knowledge and nature of training received by the case worker supporting the CALHIV household. All these variables were examined for their association as explanatory (independent) variables for viral suppression among CALHIV. The dependent variable was viral load test results of a CALHIV defined using a binary indicator–suppressed or unsuppressed. In line with the World Health organization (WHO) guidelines, a CALHIV was considered virally suppressed if the viral load count was below 1000 ribonucleic acid (RNA) copies/mL [14].

## Data analysis

Descriptive statistics were obtained and included in Table 1. A bivariable analysis was conducted using a Chi square test, together with un-adjusted Odds Ratios to compare individual variables to suppression status for CALHIV. The non-significant variables were not included in the final mixed effects model. These included "caregiver HIV status," "caregiver age," and "distance to health facility." Other independent variables turned out to be significant at bivariate analysis level, such as these two caregiver characteristics: "training for care workers," and "basic community HIV management." Similarly, "child's education level" was significant at bivariate level. However, the model convergence failed hence necessitating the elimination of the three covariates from the final model. To test for the effect of the selected variables and their association with viral load suppression for the CALHIV, a multivariate mixed effect logistic regression model was used, using *glmer* function in the LME4 package in R statistical software [15]. Statistical significance was considered at p-values of < 0.05. All statistical analyses were performed in R.

The model selection statistics including Akaike's Information Criteria (AIC), Bayesian Information Criteria (BIC) and deviance were used to validate the multivariable final model. A generalised mixed effects model was fit to the data. The full model is defined by all the independent variables selected from Tables 1 and 2, for inclusion in the final model, together with the random effect of country. The null model however had only the random effect of country. The full model (AIC = 24,922.2), the null model (AIC = 25,033) and the final model (AIC = 24918.7), indicating that the final model outperformed the full model and the null model. Missing data were not imputed. The data analysed in this study were part of routine Health records. Data for about 65,816 CALHIV were extracted with 7,918 (12%) from urban counties and 57,898 (88%) from rural counties. After removal or rows of data with missing values, a total sample size of N = 31,291 was retained for data analysis. This was considered large enough for complete case analysis. The analysis model applied was the generalised mixed effects model in R (glimer).

## Results

### Descriptive analysis results for CALHIV

A total of 31,291 CALHIV with documented viral results were included in the analysis–of which 16,307 (52.1%) are female. All CALHIV included in this analysis were actively receiving

**Table 1. Socio-demographic characteristics of CALHIV by VL suppression status.**

| Variable | | Not Suppressed | Suppressed | $X^2$ test p-value | Total |
|---|---|---|---|---|---|
| **Child's age (years)** | | | | <0.001 | |
| | 0–4 | 368 (19.8%) | 1,494 (80.2%) | | 1,862 |
| | 5–9 | 1,012 (14.0%) | 6,236 (86.0%) | | 7,248 |
| | 10–14 | 1,699 (13.2%) | 11,143 (86.8%) | | 12,842 |
| | 15–17 | 1,073 (14.4%) | 6,378 (85.6%) | | 7,451 |
| | 18+ | 252 (13.4%) | 1,636 (86.6%) | | 1,888 |
| **Sex of CALHIV** | | | | <0.001 | |
| | Female | 2,163 (13.3%) | 14,144 (86.7%) | | 16,307 |
| | Male | 2,241 (15.0%) | 12,743 (85.0%) | | 14,984 |
| **Education level** | | | | <0.001 | |
| | ECDE | 364 (16.0%) | 1,971 (84.0%) | | 2,335 |
| | Primary | 3,036 (14.0%) | 19,199 (86.0%) | | 22,235 |
| | Secondary | 593 (13.0%) | 4,104 (87.0%) | | 4,697 |
| | Tertiary | 16 (20%) | 66 (80.0%) | | 82 |
| | Not in school | 395 (20.0%) | 1,547 (80.0%) | | 1,942 |
| **Education status** | | | | <0.001 | |
| | Active | 4,009 (13.0%) | 25,342 (87.0%) | | 29,351 |
| | Inactive | 395 (20.0%) | 1,545 (80.0%) | | 1,940 |
| **Child PSS membership** | | | | <0.001 | |
| | No | 988 (16.0%) | 4,881 (84%) | | 5,869 |
| | Yes | 3,416 (13.0%) | 22,006 (87.0%) | | 25,422 |
| **Distance to facility (Km)** | | | | 0.146 | |
| | 0–4 | 2,064 (13%) | 12,926 (87.0%) | | 14,990 |
| | 5–9 | 1,735 (15.0%) | 10,176 (85.0%) | | 11, 911 |
| | 10+ | 605 (14.0%) | 3,785 (86.0%) | | 4,390 |
| **Total** | | **4,404 (14.1%)** | **26,887 (85.9%)** | | **31,291** |

CALHIV: Children and Adolescents Living With HIV

ECDE: Early Childhood Development Education

HIV: Human Immunodeficiency Virus

PSS: psycho-social support Group.

ART treatment services from the USAID PEPFAR OVC supported programs in all 34 counties in Kenya between October 2019 and September 2020. Of these, 4,404 (14.1%) had unsuppressed VL compared to 26,887 (85.9%) that had suppressed VL. Table 1 shows the social-demographic characteristics of CALHIV by VL suppression status. CALHIV aged 1–4 years appear to be the least suppressed at 80%. No significant difference is noted in VL suppression levels for other age groups, 5 years and above, that have a VL suppression averaging 86%. Female CALHIV have a higher proportion of viral suppression at 86.7% compared to their male counterparts at 85%. In relation to education status, CALHIV in tertiary institutions and those not in school are the least suppressed at 80% compared to those in secondary or primary school at 87% and 86% respectively. CALHIV actively in school or in an education institution during the year have a higher proportion of viral suppression at 87% compared to those who were not actively engaged at (80%). CALHIV registered into psychosocial support (PSS) groups had a higher proportion with suppressed VL at 87% compared to 84% who were not registered into PSS. No major differences are noted in proportions of CALHIV with suppressed VL with respect to their distance to the nearest health facility.

**Table 2. Distribution of CALHIV across levels of caregiver characteristics.**

| Variable | | Not Suppressed (n = 4,404) | Suppressed (n = 26,887) | X² test p-value | Total (31291) |
|---|---|---|---|---|---|
| **Caregiver's age (years)** | | | | 0.860 | |
| | Youth (10–35) | 864 (14.0%) | 5263 (86.0%) | | 6,127 |
| | Adult (36–65) | 3229 (14.0%) | 19663 (86.0%) | | 22,892 |
| | Elderly (66+) | 311 (14.0%) | 1961 (86.0%) | | 2,272 |
| **Caregiver's sex** | | | | 0.009 | |
| | Female | 3748 (14.0%) | 1499 (81.0%) | | 27,026 |
| | Male | 656 (15.0%) | 13169 (86.0%) | | 4,265 |
| **Caregiver education level** | | | | <0.001 | |
| | None | 362 (19.0%) | | | |
| | ECDE/Primary | 2109 (14.0%) | | | 1,861 |
| | Secondary | 940 (13.0%) | 6215 (87.0%) | | 7,155 |
| | Tertiary | 55 (11.0%) | 441 (89.0%) | | 496 |
| | Unknown | 938 (14.0%) | 5563 (86.0%) | | 6,501 |
| **Caregiver HIV status** | | | | 0.138 | |
| | Negative | 1365 (14.0%) | 7958 (86.0%) | | 9,323 |
| | Positive | 2854 (14.0%) | 17834 (86.0%) | | 20,688 |
| | Unknown | 185 (14%) | 1095 (86.0%) | | 1,280 |
| **Caregiver PSS membership** | | | | <0.001 | |
| | No | 1447 (15.0%) | 7749 (85.0%) | | 9,196 |
| | Yes | 2844 (14.0%) | 18175 (86.0%) | | 21,019 |
| **Caregiver household vulnerability status** | | | | 0.005 | |
| | Low | 536 (14.0%) | 3372 (86.0%) | | 3,908 |
| | Medium | 2663 (14.0%) | 16774 (86.0%) | | 19,437 |
| | High | 1205 (15.0%) | 6741 (85.0%) | | 7,946 |
| **Membership in VSLA** | | | | <0.001 | |
| | No | 1903 (17.0%) | 9474 (83.0%) | | 11,377 |
| | Yes | 2501 (13.0%) | 17,413 (87.0%) | | 19,914 |
| **CPARA done** | | | | 0.006 | |
| | 159 (17.0%) | 763 (83.0%) | | 922 | 11,377 |
| | 4245 (14.0%) | 26124 (86.0%) | | 30,369 | 19,914 |
| **Case Management Training for Case Worker** | | | | 0.006 | |
| | Not done | | | 0.012 | |
| | In progress | 196 (15.0%) | 1088 (85.0%) | | 1,284 |
| | Completed | 62 (10.0%) | 544 (90.0%) | | 606 |
| **Basic Community HIV Management Training** | | | | 0.003 | |
| | Not done | 726 (14.0%) | 4546 (86.0%) | | 5,272 |
| | In progress | 602 (14.0%) | 3191 (86.0%) | | 3,793 |
| | Completed | 3076 (14.0%) | 19150 (86.0%) | | 22,226 |
| **Positive Parenting Training for Case Worker** | | | | <0.001 | |
| | Not done | 1537 (15.0%) | 8078 (85.0%) | | 9,615 |
| | In progress | 428 (13.0%) | 3083 (87.0%) | | 3,511 |

(*Continued*)

**Table 2.** (Continued)

| Variable | | Not Suppressed (n = 4,404) | Suppressed (n = 26,887) | X² test p-value | Total (31291) |
|---|---|---|---|---|---|
| | Completed | 2439 (13.0%) | 15726 (87.0%) | | 18,165 |

**Note**: **CALHIV** = Children and Adolescents Living with HIV

**CPARA** = Case Plan Achievement Readiness Assessment

**ECDE** = Early Childhood Development Education

**HIV** = Human Immunodeficiency Virus

**PSS** = Psycho-social support group

**VSLA** = Voluntary Savings and Lending Associations

### Descriptive analysis results for caregivers

**Caregiver demographics.**    Data on CALHIV caregiver socio-demographic characteristics of CALHIV are shown in Table 2. Although the majority (73.2%) of CALHIV are under the care of adult caregivers aged between 36 to 65 years, there seems to be no difference in proportions of CALHIV who are virally suppressed across the age categories. The proportion of those virally suppressed stood at 86% across for all youthful, older, and elderly caregivers. Similarly, 27,026 (86%) of CALHIV were under the care of female caregivers. Similar proportions of CALHIV with VL suppression at 86% and 85% were noted for those under female and male caregivers, respectively. However, CALHIV under caregivers with tertiary level of education had a higher proportion of virally suppressed CALHIV (89%) when compared to those under the care of caregivers with no formal education at 81%. While 66% (20,688) of CALHIV were under care of HIV positive caregivers, no significant differences in proportions of CALHIV with suppressed VL with respect to caregiver's HIV status were noted. CALHIV under the care of caregivers living in households with high vulnerability had a lower proportion of virally suppressed CALHIV (85%) compared to those in households with low-moderate vulnerability (86%).

**Support mechanisms for caregivers.**    Table 2 shows characteristics of caregivers in relation to available psychosocial (PSS) and economic support mechanisms. CALHIV were generally drawn from households whose caregivers were members of a PSS group, medium household vulnerability status and enrolled in a VLSA. At least 67.2% (21,019) of CALHIV were under the care of caregivers who belonged to a psychosocial support (PSS) group. Membership of caregivers to a PSS group did not significantly influence CALHIV viral load suppression status. Viral suppression for CALHIV whose caregivers were in a PSS group stood at 86% compared to those who were not in a PSS group 85%. In contrast, CALHIV whose caregivers who belonged to a Voluntary Savings and Lending Associations (VSLA) had a higher proportion of virally suppressed CALHIV (87%) compared to those who were not (83%). Over 97% (30,369) of CALHIV come from households with completed case plans. Households with completed case plans contributed a higher proportion of CALHIV with suppressed VL (86%) as compared to those from households with no case plans (83%).

About 96% (30,007) of CALHIV were served with Caseworkers who were undergoing or had completed Case Management Training. The proportions of CALHIV with suppressed VL was evidently higher at 90% and 86% for those undergoing or have completed the training respectively when compared to those whose caseworkers had not taken the training at 85%. 83% (26,019) of CALHIV caregivers were either undergoing or had completed a Basic Community HIV Management Training. However, comparison of proportions of virally suppressed CALHIV based on caseworker completion of the Basic Community HIV Management Training did not reveal any differences. About 70% (21,676) of CALHIV caseworkers were

either undergoing or had completed a Positive Parenting Training- and the proportion of CALHIV with suppressed VL under these caregivers were higher (87%) when compared to those whose caregivers had not done the Positive Parenting Training (85%).

## Multivariable analysis results

Unadjusted and adjusted associations between CALHIV characteristics, their caregiver characteristics and VL suppression among CALHIV are presented in Table 3. Variables found significant for VL suppression using bivariate test statistics included: age group, sex, education status, and child psycho-social support group (PSS) membership. These variables were all included in the final multivariable model, except for "child's education level" that returned very large standard errors and was therefore eliminated from the final model. Following bivariate analysis, the following caregiver characteristics were included in the final model: caregiver sex, caregiver education level, caregiver PSS group membership, caregiver household vulnerability status, caregiver membership in Voluntary Savings and Lending Associations (VSLA), positive parenting for case worker and whether Case Plan Achievement Readiness Assessment (CPARA) was done. Owing to model non-convergence due to data multicollinearity with other trainings offered to the case workers, two independent variables among the caregiver characteristics were eliminated from the multivariable model: "Basic community HIV management training" and "case management training for case worker".

Three CALHIV characteristics turned out to be significantly associated with VL suppression. First, being male was associated with reduced odds of being virally suppressed (aOR = 0.88, 95%CI: 0.83, 0.94). Second, being in active education status as opposed to inactive education status was associated with increased odds of being virally Suppressed (aOR = 1.33, 95% CI: 1.07, 1.65). Lastly, child membership in PSS was associated with increased odds of VL suppression as opposed to non-membership in PSS (aOR = 1.26, 95% CI: 1.15, 1.38). Two caregiver-related factors were associated with VL suppression. First, having a male caregiver was associated with reduced odds of VL suppression (OR = 0.91, 95% CI: 0.83, 0.997). Second, caregiver membership in VSLA was associated with elevated odds of VL suppression (OR = 1.23, 95% CI: 1.13, 1.33).

## Discussion and conclusions

This study aimed at identifying CALHIV and caregiver characteristics that influence VL suppression. Prior studies have explored CALHIV characteristics that may determine VL suppression. Whereas previous studies have shown that age of CALHIV has a significant influence on VL suppression [6, 16], in this study, there is no significant difference between VL suppression levels in all age-groups in comparison to the reference level of 1–4 years. This finding is similar to those from a systematic review and meta-analysis examining cognitive and educational interventions for OVC affected by HIV/AIDS by mention authors [17]. In our study, we found that interventions targeting educational outcomes had a significant effect on VL suppression. That child membership in psychosocial support (PSS) groups was associated with increased odds of VL suppression was found in a prior study (mention study). A South African study examining psycho-educational and social interventions provided for OVC at a community-based organisation showed that psycho-educational and social interventions may improve the lives of OVC in general [18].

Several studies have considered characteristics of caregivers that may influence successful HIV care, including linkage to care, viral suppression among other health factors [9, 16, 19, 20]. ART use among caregivers has been shown to have a significant influence on viral suppression among CALHIV [18]. Food insufficiency among caregivers may also play a major

**Table 3. Factors associated with viral load suppression among orphans and vulnerable children and adolescents living with HIV in Kenya.**

|  | Unadjusted OR (95% CI) | Adjusted OR (95% CI) |
|---|---|---|
| Intercept |  | 3.72 (2.52, 5.49) |
| **Age group (Ref: 0–4 years)** |  |  |
| 5–9 | 1.51 (1.32, 1.73) | 1.16 (0.93, 1.45) |
| 10–14 | 1.59 (1.39, 1.80) | 1.12 (0.89, 1.41) |
| 15–17 | 1.42 (1.25, 1.63) | 1.00 (0.79, 1.27) |
| 18+ | 1.69 (1.41, 2.02) | 1.17 (0.90, 1.52) |
| **Sex (Ref: Female)** |  |  |
| Sex Male | 0.88 (0.83, 0.94) | 0.88 (0.83, 0.94) |
| **Education level (Ref: Not in School)** |  |  |
| ECDE | 1.37 (1.17, 1.61) |  |
| Primary | 1.56 (1.39, 1.76) |  |
| Secondary | 1.57 (1.36, 1.81) |  |
| Tertiary | 1.13 (0.65, 1.96) |  |
| **Education Status (Ref: Inactive)** |  |  |
| Active | 1.55 (1.37, 1.74) | 1.33 (1.07, 1.65) |
| **PSS Membership (Ref: No)** |  |  |
| Child PSS Membership (Yes) | 1.29 (1.19, 1.39) | 1.26 (1.15, 1.38) |
| **Distance to Facility (Ref: 0–4 km)** |  |  |
| 5–10 km | 0.96 (0.89, 1.03) |  |
| (10+ km | 1.04 (0.94, 1.16) |  |
| **Caregiver Age Category (Ref: Adolescent youth)** |  |  |
| Adult (<66 years) | 1.02 (0.94, 1.11) |  |
| Elderly (> = 66 years) | 1.03 (0.89, 1.19) |  |
| **Caregiver Sex (Ref: Female)** |  |  |
| Male | 0.91 (0.83, 0.99) | 0.91 (0.83, 0.997) |
| **Caregiver Education Level (Ref: None)** |  |  |
| ECDE and primary | 1.16 (0.98, 1.37) | 1.159 (0.98, 1.37) |
| Secondary | 1.09 (0.91, 1.30) | 1.088 (0.91, 1.30) |
| Tertiary | 1.36 (0.98, 1.89) | 1.332 (0.96, 1.86) |
| Unknown/Unspecified | 1.12 (0.93, 1.35) | 1.144 (0.95, 1.38) |
| **Caregiver HIV Status (Ref: Negative)** |  |  |
| Positive | 0.99 (0.92, 1.07) |  |
| Unknown | 1.07 (0.89, 1.29) |  |
| **Caregiver PSS Membership (Ref: No)** |  |  |
| Not applicable | 1.15 (0.90, 1.47) | 1.15 (0.90, 1.47) |
| Yes | 1.03 (0.95, 1.11) | 1.01 (0.93, 1.10) |
| **Caregiver Household Vulnerability Status (Ref: Low)** |  |  |
| High | 0.81 (0.71, 0.92) | 0.81 (0.71, 0.92) |
| Medium | 0.98 (0.89, 1.09) | 0.99 (0.89, 1.10) |
| **Caregiver Membership in VSLA (Ref: No)** |  |  |
| Yes | 1.23 (1.14, 1.33) | 1.23 (1.13, 1.33) |
| **CPARA Done (Ref: No)** |  |  |
| Yes | 0.83 (0.64, 1.08) | 0.77 (0.60, 1.03) |
| **Training for Case Workers (Ref: Not done)** |  |  |
| Completed | 0.93 (0.79, 1.11) |  |
| In progress | 1.04 (0.75, 1.45) |  |

(*Continued*)

**Table 3.** (Continued)

|  | Unadjusted OR (95% CI) | Adjusted OR (95% CI) |
|---|---|---|
| **Basic Community HIV Management Training (Ref: Not done)** |  |  |
| Completed | 0.99 (0.91, 1.10) |  |
| In progress | 0.93 (0.79, 1.02) |  |
| **Positive Parenting for Case Worker (Ref: Not done)** |  |  |
| Completed | 1.06 (0.96, 1.16) | 1.04 (0.95, 1.14) |
| In progress | 0.97 (0.84, 1.11) | 0.95 (0.82, 1.09) |

CALHIV: Children and Adolescents Living With HIV; CPARA: Case Plan Achievement Readiness Assessment; ECDE: Early Childhood Development Education; HIV: Human Immunodeficiency Virus; PSS: psycho-social support Group; VSLA: Voluntary Savings and Lending Associations.

role in viral suppression among CALHIV [18]. Having a male caregiver is associated with reduced odds of VL suppression. This finding corresponds with findings from a Tanzanian study that examined the association between caregiver sex and HIV infection among OVC. OVC with male caregivers had 40% higher likelihood to be HIV positive than those with female caregivers. The study pointed out two key aspects related to this finding. First, from a cultural standpoint, orphaned children living with male caregivers has an increased likelihood that the children are HIV-infected, and this has implications for risk assessment and referral to HIV testing services where the existing case-finding modalities, such as index testing services, are offered to individuals who are at risk of HIV exposure from the original client, who is often the mother. This approach increases the likelihood of an OVC missing out on HIV testing services in cases where the mother is dead or unreachable. Secondly, caring for others has traditionally been considered the responsibility of women and girls, particularly in the African setting. Therefore, men are more likely to provide inadequate or suboptimal care due to factors such as a lack of experience in child-rearing activities [20].

The association between social interventions that provide economic support for their families, such as caregiver membership in Voluntary Savings and Lending Associations (VLSA), and elevated odds of VL suppression in our study is similar to findings from an earlier study in Soweto, South Africa [18]. In this study, the intervention involved CBO empowering families of OVC by providing income-generating activities. Similarly, through provision of opportunities for saving and lending, caregivers of OVC are likely to be more supportive in different ways that improve the OVC's wellbeing including VL suppression. Empowering caregivers of children living with HIV to suppress is a protective factor and hence expanding such support and being more targeted will yield better outcomes.

Psychosocial Support (PSS) is central to quality care provision. Providing care for a child infected with the human immunodeficiency virus (HIV) is difficult for the child's caregiver and has an impact on the entire family system. Findings from other studies, like the one that was conducted in Botswana, show that older and young female caregivers have reported feeling overwhelmed with the demands of caregiving. They further reported psychological and emotional difficulties including feeling exhausted, depressed, and often neglected to attend to their own health [21]. In Uganda, elderly caregivers over 50 years and above have reported caregiving burdens including drastic disruptions of living arrangements, prolonged travels and absences from their homes. Other studies have reported on the mental health distress of caregivers of HIV and AIDS in the Niger Delta region of Nigeria where caregivers were found to be experiencing stress, anxiety, depression, and suicidal tendencies [22]. A study that looked at the quality of life and coping styles of caregivers of children with HIV/AIDS found that

while caregivers help children with HIV/AIDS, they also need physical and emotional support. Nurses must be aware of the importance of intervening for these caregivers to improve their coping mechanisms, reduce stress, and thus improve their quality of life [23]. According to research, social support has the potential to reduce caregiver stress and facilitate caregiver coping while time spent caring for the HIV-positive child was significantly inversely related to all aspects of quality of life [24]. UNICEF, in collaboration with governments and partners, supported a mixed method study that included literature review, assessment of laboratory data in Malawi, Uganda and Zimbabwe and interviews with health workers and caregivers in Malawi to find out what is behind these low rates. The study found that when children and their caregivers receive support from their families, communities, and health professionals, they are more likely to achieve viral load suppression [25]. This is because support networks are critical in instilling confidence in caregivers to share information about the child's HIV status and request assistance [21]. Therefore, interventions are needed to help support caregivers of children with HIV/AIDS [26].

## Limitation of the study

We acknowledge that our study is not without limitations. First, our results have limited generalizability to other settings as we conducted the study among OVC living HIV in Kenya which has recently been reclassified to a lower middle-income country (LMIC). Since we selected OVC who had documented VL results, OVC with undocumented results were excluded. Secondly, the cross-sectional nature of our data set limits causal inference from our findings since they provide data from a fixed time point. Thirdly, community- or societal-level factors were not explored in this study. Despite these limitations, the present analyses provide valuable insights on potential determinants of VL suppression among OVC living with HIV Kenya.

In conclusion, this study adds to the current knowledge on determinants of VL suppression among OVC living with HIV in Kenya. CALHIV characteristics such as child's sex, child's education status (whether currently active in school or inactive) and child's membership in a psychosocial support group are key determinants of VL suppression. Similarly, caregiver sex and membership in a voluntary savings and lending association also influence VL suppression. Programs are encouraged to target these individual and caregiver-related factors as they are fundamental to VL suppression among OVC living with HIV. This study may inform policy development and advocacy to improve viral load suppression and health outcomes among orphans and vulnerable children living with HIV.

## Acknowledgments

The authors would like to thank USAID for making the data set for analysis available and granting us permission to engage in this publication process. We also acknowledge colleagues from the OVC implementing partners monitoring and evaluation teams for submitting the data. Finally, we wish to thank APHRC staff who conducted training on data analysis and scientific writing.

## Author Contributions

**Conceptualization:** Brian Onyango, Rose Mokaya, Jeniffer Wasianga, Hesborn Wao, Dunstan Achwoka, Nelson Onyango, Damazo T. Kadengye.

**Data curation:** Brian Onyango, Hesborn Wao, Dunstan Achwoka, Nelson Onyango, Damazo T. Kadengye.

**Formal analysis:** Brian Onyango, Hesborn Wao, Nelson Onyango, Damazo T. Kadengye.

**Funding acquisition:** Brian Onyango.

**Investigation:** Brian Onyango, Hesborn Wao, Damazo T. Kadengye.

**Methodology:** Brian Onyango, Rose Mokaya, Jeniffer Wasianga, Hesborn Wao, Dunstan Achwoka, Damazo T. Kadengye.

**Project administration:** Brian Onyango, Rose Mokaya, Hesborn Wao, Damazo T. Kadengye.

**Resources:** Brian Onyango, Rose Mokaya, Hesborn Wao, Damazo T. Kadengye.

**Software:** Brian Onyango, Hesborn Wao, Nelson Onyango, Damazo T. Kadengye.

**Supervision:** Hesborn Wao, Nelson Onyango, Damazo T. Kadengye.

**Validation:** Brian Onyango, Rose Mokaya, Jeniffer Wasianga, Hesborn Wao, Dunstan Achwoka, Nelson Onyango, Damazo T. Kadengye.

**Visualization:** Brian Onyango, Hesborn Wao, Nelson Onyango, Damazo T. Kadengye.

**Writing – original draft:** Brian Onyango, Rose Mokaya, Jeniffer Wasianga, Hesborn Wao, Dunstan Achwoka, Damazo T. Kadengye.

**Writing – review & editing:** Brian Onyango, Rose Mokaya, Jeniffer Wasianga, Hesborn Wao, Dunstan Achwoka, Damazo T. Kadengye.

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
