## [Decision Letter · Decision Letter 0]

10 Aug 2022

PGPH-D-22-00954

Factors associated with viral load suppression among orphans and vulnerable children and adolescents living with HIV in Kenya

Dear Dr. Onyango,

Thank you for submitting your manuscript to PLOS Global Public Health. After careful consideration, we feel that it has merit but does not fully meet PLOS Global Public Health’s publication criteria as it currently stands. Therefore, we invite you to submit a revised version of the manuscript that addresses the points raised during the review process.

Two reviewers have assessed your manuscript; please address all of their comments thoroughly in your revisions and response. We are issuing a decision on your manuscript at this point to prevent further delays in the evaluation of your manuscript. Please be aware that the editor who handles your revised manuscript might find it necessary to invite additional reviewers to assess this work once the revised manuscript is submitted. However, we will aim to proceed on the basis of these reviews if possible.

We look forward to receiving your revised manuscript.

Kind regards,

Julia Robinson

Executive Editor

Journal Requirements:

1. Please update your online Competing Interests statement. If you have no competing interests to declare, please state: “The authors have declared that no competing interests exist.”

2. Please amend your detailed online Financial Disclosure statement. This is published with the article. It must therefore be completed in full sentences and contain the exact wording you wish to be published.

Please state what role the funders took in the study. If the funders had no role in your study, please state: “The funders had no role in study design, data collection and analysis, decision to publish, or preparation of the manuscript.”

3. In the Funding Information you indicated that no funding was received. Please revise the Funding Information field to reflect funding received.

Please ensure that the funders and grant numbers match between the Financial Disclosure field and the Funding Information tab in your submission form. Note that the funders must be provided in the same order in both places as well.

Additional Editor Comments (if provided):

Reviewers' comments:

Reviewer's Responses to Questions

**Comments to the Author**

1. Does this manuscript meet PLOS Global Public Health’s publication criteria? Is the manuscript technically sound, and do the data support the conclusions? The manuscript must describe methodologically and ethically rigorous research with conclusions that are appropriately drawn based on the data presented.

Reviewer #1: Partly

Reviewer #2: Partly

2. Has the statistical analysis been performed appropriately and rigorously?

Reviewer #1: Yes

Reviewer #2: No

3. Have the authors made all data underlying the findings in their manuscript fully available (please refer to the Data Availability Statement at the start of the manuscript PDF file)?

Reviewer #1: Yes

Reviewer #2: Yes

4. Is the manuscript presented in an intelligible fashion and written in standard English?

Reviewer #1: No

Reviewer #2: Yes

5. Review Comments to the Author

Reviewer #1: Manuscript needs significant copy editing. There were many typos and grammatical errors.

This reviewer was also concerned by the lack of IRB review. Best practice is always to include IRB review, even for secondary data analysis. This is a non-negotiable requirement when working on PEPFAR (through CDC) awards and represents a basic tenet of Good Clinical Practice in research.

Reviewer #2: The attempt to characterize factors associated with virologic suppression in OVC-CALHIV is commendable. However, there is need to clarify some aspects of the manuscript in its present form:

1. Overall Objective: the authors need to make clear what outcome is being assessed-is it virologic suppression OR non-suppression? The objective stated in the introduction section is at variance with the line of discussion of results.

2. In the results section and reflected in Tables 1 and 2: the authors should show and document if there are any differences between those with suppressed vs. unsuppressed viral load across the variables that are listed in the 2 Tables (show results of Chi-square tests or other tests conducted). This will also allow assessment of which variables were appropriate to include in the Model.

3. Table 1: the age categories should reworked to conform with either WHO format (0-9, 10-14, 15-18 OR PEPFAR fine age bands-0-4, 5-9, 10-14, 15-18/19 or 18+). This may modify the impact of age on the primary outcome of interest.

-there are non-standard abbreviations used in the Tables-these should be spelt out as footnotes to the Tables respectively.

4. Missing Data: Can the authors explain how they handled missing data in the course of analysis of program level data? were there any sensitivity analyses conducted to handle this limitation? OR were there no missing data elements for any of the variables extracted?

5. Abstract: The aim stated in the abstract is to characterize SUB-OPTIMAL virologic suppression as opposed o VLS< 1000c/ml stated in the main text of the manuscript-please harmonize. Additionally, please delete the "ref level" in the body of the abstract and just include only the aOR's and 95% CIs. Endeavor to define all variables fully in the Methods section and reserve the abstract for key findings only.

6. PLOS authors have the option to publish the peer review history of their article (what does this mean?). If published, this will include your full peer review and any attached files.

**Do you want your identity to be public for this peer review?** For information about this choice, including consent withdrawal, please see our Privacy Policy.

Reviewer #1: No

Reviewer #2: No

---

## [Decision Letter · Decision Letter 1]

28 Nov 2022

PGPH-D-22-00954R1

Factors associated with viral load suppression among orphans and vulnerable children and adolescents living with HIV in Kenya

Dear Dr. Brian Otieno Onyango

Thank you for submitting your manuscript to PLOS Global Public Health. After careful consideration, we feel that it has merit but does not fully meet PLOS Global Public Health’s publication criteria as it currently stands. Therefore, we invite you to submit a revised version of the manuscript that addresses the points raised during the review process.

We look forward to receiving your revised manuscript.

Kind regards,

Claudia P. Cortes, MD

Academic Editor

Journal Requirements:

Additional Editor Comments (if provided):

thank you very much for sending a new version of your work.

while most of the observations from the first round of review have been clarified, the IRB's authorization to conduct this work remains to be clarified. that is a fundamental issue that has not been fully resolved.

The matter regarding IRB approval has not been effectively addressed by the authors. The implications of this situation are made considerably worrysome by the fact that the principal author is a representative of the US government. The use (and release) of secondary data necessitates independent ethics clearance by a party other than the institutional owner of the data (USAID or the Kenyan government). All authors should be held accountable for it because it is a fundamental principle of good clinical practice.

Also a new reviewer has incorporated comments and observations that deserve consideration.

Reviewers' comments:

Reviewer's Responses to Questions

**Comments to the Author**

1. If the authors have adequately addressed your comments raised in a previous round of review and you feel that this manuscript is now acceptable for publication, you may indicate that here to bypass the “Comments to the Author” section, enter your conflict of interest statement in the “Confidential to Editor” section, and submit your "Accept" recommendation.

Reviewer #1: All comments have been addressed

Reviewer #2: (No Response)

Reviewer #3: (No Response)

2. Does this manuscript meet PLOS Global Public Health’s publication criteria? Is the manuscript technically sound, and do the data support the conclusions? The manuscript must describe methodologically and ethically rigorous research with conclusions that are appropriately drawn based on the data presented.

Reviewer #1: Partly

Reviewer #2: Yes

Reviewer #3: Partly

3. Has the statistical analysis been performed appropriately and rigorously?

Reviewer #1: Yes

Reviewer #2: Yes

Reviewer #3: Yes

4. Have the authors made all data underlying the findings in their manuscript fully available (please refer to the Data Availability Statement at the start of the manuscript PDF file)?

Reviewer #1: Yes

Reviewer #2: Yes

Reviewer #3: (No Response)

5. Is the manuscript presented in an intelligible fashion and written in standard English?

Reviewer #1: Yes

Reviewer #2: Yes

Reviewer #3: No

6. Review Comments to the Author

Reviewer #1: (No Response)

Reviewer #2: Consider re-casting the results and conclusion sections of the abstract to read as follows (detailed below):

Results: Determinants of VL suppression included child’s education status (aOR = 1.33; 95% CI: 1.07, 1.65), membership of a psychosocial support group (aOR = 1.258; 95% CI: 1.15, 1.38), and membership of a voluntary savings and lending association (VSLA) (aOR = 1.226; 95% CI: 1.129, 1.33). In addition, child’s sex (aOR=0.88; 95% CI: 0.83, 0.94), caregiver sex (aOR=0.909; 95% CI: 0.839, 0.997) and “high” status for caregiver household vulnerability (aOR = 0.81; 95% CI: 0.71, 0.924), had an inverse relationship with VL suppression.

Conclusion: CALHIV characteristics including child’s sex, child’s education status (whether currently active in school or inactive) and child’s membership in a psychosocial support group were key determinants of VL suppression. Similarly, caregiver sex and membership in a voluntary savings and lending association also influence VL suppression.

Additionally, the proportions of sites that were rural and urban are currently missing (underlined in red in the attached PDF)

-to be included

Reviewer #3: This is an important article on a very important topic. There are key areas that are needed to be addressed to move forward. The goal is to move the field forward and currently the analysis is without considerable thought to these areas.

1. The impact of of the constructs examined in the analysis are not well developed. The authors note that "Several reasons cited for poor adherence to ART include financial instability, poverty and food insecurity, limited access to medical resources, and costs of transportation. Good adherence and retention are noted to be necessary requirements for achieving viral suppression. Adolescents are particularly vulnerable to household poverty, yet adherence to treatment regimens requires a level of economic stability that many poor youths in Africa do not have/experience [28][29]." However, do not go into detail as to how this may impact viral suppression. What is the conceptual framework underpinning the assumption that economic stability might be protective and helpful. There are multiple frameworks that could be described to set the article up in the introduction section better.

2. Also, there are other factors that are examined that relate potentially to structural barriers to care - ie, distance to the facility, adequate training, that are included in the analysis but not introduced from a theoretical perspective. Why might these be helpful in exploring.

3. The authors provide information in the results that should be included in the methods. For example the description and handling of missing data should be moved to the methods section.

4. The authors describe that the model was initially not stable and did not converge. "Owing to model non-convergence, two independent variables among the caregiver characteristics were eliminated from the multivariable model: “Basic community HIV management training” and “case management training for case worker.” These variables were removed. Were these thought to be correlated? If so, please provide a correlation coefficient. If not, the authors need to justify why these variables were removed. Currently, it looks like it was randomly determined which is not a thoughtful approach.

5. The focus of the article is only included in 2 sentences in the discussion. "Voluntary Savings and Lending Associations (VLSA), and elevated odds of VL suppression in our study is similar to findings from an earlier study in Soweto, South Africa. [21]. In this study, the intervention involved CBO empowering families of OVC by providing income generating activities. Similarly, through provision of opportunities for saving and lending,

caregivers of OVC are likely to be more supportive in different ways that improve the OVC’s

wellbeing including VL suppression." This seems unexplored. More could be said about this. How do the authors suppose that this is impacting viral suppress? Is it providing more access? Is there a protective mechanism at play. If the introduction had been set up better then this would provide evidence of answering these questions. It is also important for the authors to include information about the implications of the findings.

6. The authors comment that male caregivers are associated with lower viral suppression, but do not provide an explanation or how to better address this. Male caregivers can't be removed. How do the authors suggest they be supported. What evidence is suggested as a cause for this relationship - are we devoting more support for females and leaving males out? More meat is needed here.

7. The authors state the inability to look at key community and social factors but describe many causes and examine factors at these levels. There are societal and community level factors explored through economic advancement (voluntary savings) and social support. I don't see that PSS is commented on anywhere in the manuscript, which is a big oversight.

7. PLOS authors have the option to publish the peer review history of their article (what does this mean?). If published, this will include your full peer review and any attached files.

**Do you want your identity to be public for this peer review?** For information about this choice, including consent withdrawal, please see our Privacy Policy.

Reviewer #1: No

Reviewer #2: No

Reviewer #3: No

---

## [Editor Report · Decision Letter 2]

27 Feb 2023

Factors associated with viral load suppression among orphans and vulnerable children and adolescents living with HIV in Kenya

PGPH-D-22-00954R2

Dear Dr Brian Otieno Onyango

We are pleased to inform you that your manuscript 'Factors associated with viral load suppression among orphans and vulnerable children and adolescents living with HIV in Kenya' has been provisionally accepted for publication in PLOS Global Public Health.

Best regards,

Claudia P. Cortes, MD

Academic Editor